# Ranking DMUs by Combining Cross-Efficiency Scores Based on Shannon’s Entropy

**DOI:** 10.3390/e21050467

**Published:** 2019-05-04

**Authors:** Yueh-Chiang Lee

**Affiliations:** Department of Business Administration, Vanung University, Zhongli, Taoyuan 32061, Taiwan; yclee@mail.vnu.edu.tw

**Keywords:** data envelopment analysis, cross efficiency, Shannon’s entropy, ranking

## Abstract

Cross-efficiency evaluation is an effective approach for ranking decision-making units (DMUs), and there exist different perspectives from different cross-efficiency evaluation models. However, efficiency ranking results derived from cross-efficiency models may not be the same, and these models may provide some precious information that we cannot ignore. In this case, it may not be easy for one to decide which method should be used in some underlying assumptions, and we need several cross-efficiency evaluation models to measure simultaneously the cross-efficiency scores of DMUs. Hence, combining different viewpoints for ranking DMUs is a possible way to apply cross-efficiency evaluation. Since Shannon’s entropy is an effective tool to measure uncertainty, in this study we adopt the idea of Shannon’s entropy to combine cross-efficiency scores, which are obtained from different evaluation models, for comparison of DMUs. An example of commercial banks in Taiwan is used to illustrate the idea proposed in this paper.

## 1. Introduction

Data envelopment analysis (DEA) has been widely used as a powerful performance measurement tool to evaluate the efficiency of different comparable entities, namely decision-making units (DMUs) that consume multiple inputs to produce multiple outputs. Since each DMU measures its efficiency with the most favorable weights to itself, the nature of self-evaluation may lead to the case that many DMUs are evaluated as efficient and cannot be discriminated any further. In other words, DEA suffers from lack of discrimination power on efficient units, and the self-evaluation allows each DMU to be evaluated with its most favorable weights. The inputs and outputs will be heavily weighted for a favorable DMU, whereas those not favorable to the DMU will be less weighted. A commonly recognized problem of DEA is assessing too many units as efficient. In the literature DEA cross-efficiency evaluation has shown to be an effective methodology to improve the discrimination power among CCR efficient DMUs.

The DEA cross-efficiency method extends the self-evaluation into the peer-evaluation of conventional DEA models. The idea of cross-efficiency is to use the set of weights selected by each DMU in calculating its own efficiency as a specific set of common weights by which to calculate the efficiency of all other DMUs. The average of self- and peer-evaluation efficiency scores is treated as the overall cross-efficiency score of the DMU being evaluated. The cross-efficiency evaluation not only provides a ranking result among DMUs, but also removes unrealistic weight without the need of the elicitation of weight restrictions from experts [1]. Nevertheless, the conventional cross-efficiency evaluation still has some flaws. DEA models might exist with multiple optimum weights, which may result in different cross-efficiency scores. Under such a situation, we have different ranking results of DMUs. To tackle this problem, Sexton et al. [2] and Doyle and Green [3] introduced the secondary goals approach for the choice of weights among the alternative optimal solutions. Due to the sound property of the cross-efficiency evaluation, this approach has been applied to various industries, and a number of cross-efficiency models and applications have been reported in the literature (for examples, Liang et al. [4,5], Ramón et al. [6], Wang et al. [7], Wu et al. [8,9], Liu [10], Oukil [11], Al-Siyabi et al. [12], and Liu et al. [13]). Wu et al. [9] made a good review of the literature involving models, and Liu [10] described many novel applications.

Information entropy is an effective tool to measure the uncertainty. According to the idea of entropy, the amount or quality of information is one of the determinants for making decisions accurately. To this end, it has been widely applied to different cases of assessments, and there are also several articles that integrated the entropy and DEA models. Soleimani-Damaneh and Zarepisheh [14] employed Shannon’s entropy to integrate a family of DEA efficiency scores, which are measured from different DEA models, into an efficiency index for ranking DMUs. Xie et al. [15] used Shannon’s entropy to derive the degree of importance of each DMU. Then they merged the calculated efficiency scores and the degrees of importance to help discriminate traditional DEA models. Qi and Guo [16] proposed a modified weight-restricted DEA model for the derivation of non-zero optimal weights, where Shannon’s entropy is used to aggregate those weights to be the common weights. Wang et al. [17] used the DEA entropy model to find the cross-efficiency intervals with imprecise inputs and outputs, and all DMUs are ranked according to the distance to ideal positive cross-efficiency. Lu and Liu [18] took into account the aggressive and benevolent formulations at the same time, and consequently, a number of cross-efficiency intervals are obtained for each DMU. The entropy is then used to construct a numerical index for ranking DMUs.

Different ways for determining the weights produce different cross efficiencies, and lead to different ranking results of DMUs [9,18]. In this case, it may be not easy for one to decide which method should be used in some underlying assumptions, and we need several cross-efficiency evaluation models to measure simultaneously the cross-efficiency scores of DMUs. In particular, each of the models and viewpoints might have some precious advantages that we cannot ignore. Zeleny [19] first proposed to apply Shannon’s entropy as a coefficient of importance degree in multiple criteria decision analysis. As noted in Soleimani-Damaneh and Zarepisheh [14], if the outcomes of the performance analysis are important, one might try different models, from which to combine different results and viewpoints together. For this reason, in this study we adopt the idea of Soleimani-Damaneh and Zarepisheh [14] to calculate Shannon’s entropy of the obtained cross-efficiencies from several cross-efficiency models for ranking DMUs.

In the sections that follow, we first introduce the cross-efficiency evaluation and the associated alternative secondary goal models. Then we develop the solution procedure to combine cross-efficiency scores from different evaluation models with Shannon’s entropy for the comparison of DMUs. An example of commercial banks in Taiwan is applied to illustrate the ideal proposed in this study. Finally, some conclusions of this study are presented.

## 2. Cross-Efficiency Evaluation

Let *X_ij_* and *Y_rj_* denote the *i*-th input, *i* = 1,…, *m*, and *r*-th output, *r* = 1,…, *s*, respectively, of the *j*-th DMU, *j* = 1,…, *n*. The DEA model proposed by Charnes et al. [20] for calculating the efficiency of DMU *d* under the assumption of constant returns-to-scale, referred to as the CCR model, is:(1)$$E_{dd}\;=\;\max\;\sum_{r=1}^{s}u_{rd}Y_{rd}\;s.t.\;\sum_{i=1}^{m}v_{id}X_{id}\;=\;1\;\sum_{r=1}^{s}u_{rd}Y_{rj}-\sum_{i=1}^{m}v_{id}X_{ij}\leq 0,\;j=\;1,\text{…},\;n,\;u_{rd},\;v_{id}\geq 0,\;r=1,\text{…},\;s,\;i=\;1,\text{…},\;m,$$where $u_{rd}$ and $v_{id}$ are the weights assigned to the *s* outputs and *m* inputs, respectively.

In the cross-efficiency evaluation we use the optimal weights obtained from (1) to calculate the cross-efficiency scores. Specifically, if $v_{id}^{*}$ (*i* = 1,…, *m*) and $u_{rd}^{*}$ (*r* = 1,…, *s*) is an optimal solution of (1) for a given DMU *d*, then the cross-efficiency of DMU *j* (*j* = 1,…, *n*, j≠d) peer-evaluated by DMU *d* is given by:(2)$$E_{dj}=\frac{\sum_{r=1}^{s}u_{rd}^{*}Y_{rj}}{\sum_{i=1}^{m}v_{id}^{*}X_{ij}},\;d,\;j=\;1,\text{…},\;n$$

The cross-efficiency score of DMU *j*, *j* = 1,…, *n*, is calculated as the average of its cross-efficiencies obtained with the weights of all the DMUs. In other words, the cross-efficiency of DMU *j* is defined as:(3)$${\bar{E}}_{j}=\frac{1}{n}\sum_{d=1}^{n}E_{dj},\;j=1,\ldots ,n$$

## 3. Alternative Secondary Goal Models

The cross-efficiency scores calculated from DEA models may not be unique because of the multiple optimal solutions for DEA weights, and we obtain different efficiencies with different solutions of DEA weights. To eliminate the non-uniqueness, Sexton et al. [2] and Doyle and Green [3] proposed to use secondary goals to choose the weights among the optimal solutions. Since then a number of cross-efficiency models and applications have been reported in the literature. From the record of Web-of-Science, the first three most cited articles in cross-efficiency study are, namely Doyle and Green [3] and Liang et al. [4,5]. Therefore, in this paper we employ these three studies to measure cross-efficiency scores of DMUs, respectively, and combine their obtained results together to calculate Shannon’s entropy for comparison of DMUs.

### 3.1. The Method of Secondary Goals

The most commonly used secondary goals approach is proposed by Doyle and Green [3]. They defined the aggregate efficiency to be the weighted average of the other *n* − 1 efficiencies, with the weight of $\sum_{i=1}^{m}v_{id}X_{id}$/$\sum_{j=1,j\neq d}^{n}\sum_{i=1}^{m}v_{id}X_{ij}$ for DMU *d* to obtain the following model:(4)$$\max\;\frac{\sum_{j=1,j\neq d}^{n}\sum_{r=1}^{s}u_{rd}Y_{rd}}{\sum_{j=1,j\neq d}^{n}\sum_{i=1}^{m}v_{id}X_{id}}\;s.t.\sum_{r=1}^{s}u_{rd}Y_{rd}\;=\;E_{dd}\sum_{i=1}^{m}v_{id}X_{id}\;\sum_{r=1}^{s}u_{rd}Y_{rj}-\sum_{i=1}^{m}v_{id}X_{ij}\leq 0,\;j=1,\ldots ,\;n\;u_{rd},\;v_{id}\geq 0,\;r=1,\ldots ,\;s,\;I=1,\ldots ,\;m$$ where $E_{dd}$ is the CCR efficiency of DMU *d* obtained from (1).

This model is a linear fractional program, which can be linearized by applying the variable substitution technique of Charnes and Cooper [21] as follows:(5)$$\max\;\sum_{j=1,j\neq d}^{n}\sum_{r=1}^{s}u_{rd}Y_{rj}\;s.t.\;\sum_{j=1,j\neq d}^{n}\sum_{i=1}^{m}v_{id}X_{id}\;=\;1\;\sum_{r=1}^{s}u_{rd}Y_{rd}\;=\;E_{dd}\sum_{i=1}^{m}v_{id}X_{id}\;\sum_{r=1}^{s}u_{rd}Y_{rj}-\sum_{i=1}^{m}v_{id}X_{ij}\leq 0,\;j=1,\ldots ,\;n\;u_{rd},\;v_{id}\geq 0,\;r=1,\ldots ,\;s,\;i=1,\ldots ,\;m$$

Model (5) is known as the benevolent formulation for cross-efficiency evaluation that aims to maximize the cross-efficiency scores of the other *n* − 1 DMUs to some extent. The set of weights obtained from this model may not be the same as that obtained from Model (1). However, due to the constraint of “$\sum_{r=1}^{s}u_{rd}Y_{rd}$ = $E_{dd}\sum_{i=1}^{m}v_{id}X_{id}$“, it will produce the same efficiency $E_{dd}$ for DMU *d*. When Model (5) performs *n* time, we have *n* different sets of optimal soltions $v_{id}^{*}$ and $u_{rd}^{*}$, and these optimal weights sets are used to calculate the cross-efficiency for each DMU by Equations (2) and (3).

### 3.2. The Method of Mean Absolute Deviation

Liang et al. [4] pointed out that Model (1) can be represented equivalently in the following deviation variable form:(6)$$\min\;s_{d}\;s.t.\;\sum_{r=1}^{s}v_{id}X_{id}=1\;\sum_{r=1}^{s}u_{rd}Y_{rj}-\sum_{r=1}^{s}v_{id}X_{ij}+s_{j}=0,\;j=1,\ldots ,n\;u_{rd},\;v_{id},\;s_{j}\geq 0,\;r=\;1,\text{…},\;s,\;i=\;1,\text{…},\;m,\;j=\;1,\ldots ,\;n.$$ where $s_{d}$ and $s_{j}$ are the deviation variables for DMUs *d* and *j*, respectively. Under this model, if $s_{d}^{*}$ = 0, then DMU *d* is efficient. Otherwise, the efficiency score of DMU *d* is $E_{dd}$ = $1-s_{d}^{*}$. Liang et al. [4] referred to this deviation variable $s_{j}$ as the *d*-inefficiency of DMU *j*.

To search for the minimization of the variation among DMUs, they proposed the following secondary goal model:(7)$$\min\;\frac{1}{n}\sum_{j=1}^{n}\left|s_{j}-{\bar{s}}_{j}\right|\;s.t.\;\sum_{r=1}^{s}v_{id}X_{id}=1\;\sum_{r=1}^{s}u_{rd}Y_{rd}=1-s_{d}^{*}\;\sum_{r=1}^{s}u_{rd}Y_{rj}-\sum_{r=1}^{s}v_{id}X_{ij}+s_{j}=0,\;j=1,\ldots ,n\;u_{rd},\;v_{id},\;s_{j}\geq 0,\;r=\;1,\text{…},\;s,\;i=\;1,\text{…},\;m,\;j=\;1,\ldots ,\;n.$$ where ${\bar{s}}_{j}=1/n\sum_{j=1}^{n}s_{j}$.

Let $\alpha_{j}=\frac{1}{2}(\left|s_{j}-{\bar{s}}_{j}\right|+(s_{j}-{\bar{s}}_{j}))$ and $\beta_{j}=\frac{1}{2}(\left|s_{j}-{\bar{s}}_{j}\right|-(s_{j}-{\bar{s}}_{j}))$. Model (7) can be transformed into the following linear program:(8)$$\min\;\frac{1}{n}\sum_{j=1}^{n}\left(\alpha_{j}+\beta_{j}\right)\;s.t.\;\sum_{r=1}^{s}v_{id}X_{id}=1\;\sum_{r=1}^{s}u_{rd}Y_{rd}=E_{dd}\;\sum_{r=1}^{s}u_{rd}Y_{rj}-\sum_{r=1}^{s}v_{id}X_{ij}+s_{j}=0,\;j=1,\ldots ,n\;\alpha_{j}-\beta_{j}=s_{j}-\frac{1}{n}\sum_{j=1}^{n}s_{j},\;j=1,\ldots ,n\;u_{rd},\;v_{id},\;s_{j},\;\alpha_{j},\;\beta_{j}\geq 0,\;r=\;1,\text{…},\;s,\;i=\;1,\text{…},\;m,\;j=\;1,\ldots ,\;n.$$

Similar to (1), the cross-efficiency ${\bar{E}}_{j}$ for DMU *j* is calculated through Equations (2) and (3).

### 3.3. The Method of Game Cross-Efficiency

Liang et al. [5] viewed each DMU as a player that searched for the maximization of its own efficiency, under the condition that the cross efficiency of each of the other DMUs does not deteriorate. Based on the idea of Liang et al. [5], the cross-efficiency of DMU *d* relative to DMU *j* can be attained through the following linear program:(9)$$E_{dj}\;=\;\max\;\sum_{r=1}^{s}u_{rd}Y_{rj}\;s.t.\;\sum_{i=1}^{m}v_{id}X_{ij}\;=\;1,\;\sum_{r=1}^{s}u_{rd}Y_{rd}-E_{dd}\sum_{i=1}^{m}v_{id}X_{id}\;=\;0,\;\sum_{r=1}^{s}u_{rd}Y_{rd}-\sum_{i=1}^{m}v_{id}X_{ij}\leq 0,\;j=\;1,\text{…},\;n,\;j\neq d,\;u_{rd},\;v_{id}\geq 0,\;r=\;1,\text{…},\;s,\;i=\;1,\text{…},\;m.$$

In Equation (9) DMU *j* seeks to maximize its own cross-efficiency score under the condition of maintaining the self-evaluation efficiency of DMU *d* unchanged. The optimal solution $E_{dj}^{*}$ is the cross-efficiency of DMU *j* peer-evaluated by DMU *d*, and the cross-efficiency of DMU *j* is calculated as ${\bar{E}}_{j}=\frac{1}{n}\sum_{d=1}^{n}E_{dj},\;j=1,\ldots ,n$, which is defined in Equation (3).

## 4. Combination of Cross-Efficiencies with Shannon’s Entropy

Shannon’s entropy plays a crucial role that has an important impact on information theory [22]. Due to the sound property of the entropy, it has been now widely employed to assess importance or uncertainty in theoretical and application studies. With the basis of this idea, Soleimani-Damaneh and Zarepisheh [14] proposed a solution procedure to obtain the importance degree from different DEA models, and an efficiency index is calculated by the combination of the derived efficiencies for ranking DMUs. In this study the idea of Soleimani-Damaneh and Zarepisheh [14] is adopted as a tool to calculate Shannon’s entropy for ranking DMUs with the cross-efficiency scores obtained from different evaluation models.

Let *n* and *q* denote the numbers of DMUs and cross-efficiency evaluation models, respectively, and the obtained cross-efficiency results are represented as the matrix $E_{n\times q}$. Each row of *E* stands for a DMU and each of its columns corresponds to a cross-efficiency evaluation model. In other words, $E_{j\times p}$ represents the cross- efficiency of DMU *j*, *j* = 1,…,*n*, derived by a cross-efficiency evaluation model $C_{p}$ for *p* = 1,…,*q*. With the cross-efficiency scores evaluated from different models, the approach of Shannon’s entropy for ranking DMUs can be summarized as the following algorithm.

Step 0. Calculate the CCR efficiency scores for all DMUs by using Model (1).

Step 1. Calculate the cross-efficiency matrix $E_{n\times q}$. $$E=\left[\begin{array}{cccc} E_{11} & E_{12} & \cdots & E_{1q}\\ E_{21} & E_{22} & \cdots & E_{2q}\\ E_{31} & E_{32} & \cdots & E_{3q}\\ \begin{array}{l} \;\vdots \\ E_{n1} \end{array} & \begin{array}{l} \;\vdots \\ E_{n2} \end{array} & \begin{array}{l} \cdots \end{array} & \begin{array}{l} \;\vdots \\ E_{nq} \end{array} \end{array}\right]$$

Step 2. Normalize the cross-efficiency matrix by setting $${\hat{E}}_{jp}=\frac{E_{jp}}{\sum_{j=1}^{n}E_{jp}},\;j=1,\ldots ,n,\;p=1,\ldots ,q.$$

Step 3. Compute Shannon’s entropy $H_{p}$ for each cross-efficiency evaluation model as $H_{p}$ = $-{\left(\ln n\right)}^{-1}\sum_{j=1}^{n}{\hat{E}}_{jp}\ln{\hat{E}}_{jp},\;p=1,\ldots ,q,$, where ${\left(\ln n\right)}^{-1}$ is the entropy constant.

Step 4. Set $D_{p}=1-H_{p}$ as the degree of diversification for each cross-efficiency evaluation model.

Step 5. Calculate the degree of importance for model $C_{p}$, and let $w_{p}$ = $\frac{D_{p}}{\sum_{p=1}^{q}D_{p}},$
p=1,…,q, as the weight coefficient of model $C_{p}$.

Step 6. Calculate the composite cross-efficiency scores $E_{j}^{C*}$ = $\sum_{p=1}^{q}w_{p}E_{jp},\;j=1,\ldots ,n,$ for comparison of DMUs. The larger the value of $E_{j}^{C*}$ the better the DMU is.

## 5. Example

Liu [10] investigated the cross-efficiency scores of 22 commercial banks in Taiwan. Three inputs and three outputs are considered in measuring cross-efficiencies: the labor cost ($X_{1}$), the physical capital ($X_{2}$), and the purchase funds ($X_{3}$) as the inputs, and the demand deposits ($Y_{1}$), the short-term loans ($Y_{2}$), and the medium- and long-term loans ($Y_{3}$) as the outputs. Data for the 22 banks are shown in Table 1. We use this dataset to illustrate how our model is applied to calculate the composite cross-efficiency scores with Shannon’s entropy for ranking the commercial banks. We first measure CCR efficiency scores for the 22 commercial banks, and their results are listed in the last column of Table 1.

Following Step 1 of the algorithm introduced in Section 4, we employ Models (5), (8) and (9) to calculate the cross-efficiency scores of the commercial banks, respectively, with the results shown in Table 2, Table 3 and Table 4. The elements of the cross-efficiency matrix $E_{n\times q}$, which stem from the last rows of Table 2, Table 3 and Table 4, are listed in Table 5. The number in the parenthesis of Table 5 is the ranking place of the commercial bank. It can be found in Table 5 that inconsistent ranking occurs for banks under different models. For example, Bank no. 4 has fifth place under the evaluation of Models (5) and (8), but is ranked as third place under Model (9). From the practical standpoint, the combination of the results obtained from the three models seems to be a reasonable method for comparison of the commercial banks. 

The normalized cross-efficiency scores are presented in columns 2–4 of Table 6, and Shannon’s entropies for the three models (i.e., *q* = 3) are measured as $H_{1}$ = 0.9951 (Model (5)), $H_{2}$ = 0.9952 (Model (8)), and $H_{3}$ = 0.9954 (Model(9)), respectively. According to Step 4 of the algorithm, the associated degrees of diversification for the three models are $D_{1}$ = 0.0049, $D_{2}$ = 0.0048, and $D_{3}$ = 0.0046, respectively. With these values, we can easily calculate the degrees of importance as $w_{1}$ = 0.3419, $w_{2}$ = 0.3348, and $w_{3}$ = 0.3233, respectively. The composite cross-efficiency scores are then obtained by using Step 6 of the algorithm, with the results shown in the second-to-last column of Table 6.

Of the 22 commercial banks, based on their calculated composite cross-efficiencies, bank no. 16 is in first place, followed by banks nos. 3, 11, 19, and 4 subsequently. This shows that combining the efficiency results of different cross-efficiency models gives more practical ranking results for decision makers compared with using each of cross-efficiency models respectively. For example, if a DMU has a strict maximum output value for any output item, then it may be measured as an efficient unit by some cross-efficiency evaluation models. Under this situation the model might decide the efficiency based upon one factor, and it seems to be not reasonable. This can be avoided by using the procedure for the combination of different cross-efficiencies proposed in this study.

## 6. Conclusions

Cross-efficiency evaluation is an effective approach for ranking DMUs. Nevertheless, different approaches measure cross-efficiency scores from different perspectives, and the efficiency rankings derived from evaluation models proposed in the literature may not be the same. In this case, we are hardly able to determine which method should be used. Besides, each model used for generating cross-efficiencies might have valuable viewpoints that we need to care about. To this end, this study applies the idea of Soleimani-Damaneh and Zarepisheh [14] to calculate Shannon’s entropy of the obtained cross-efficiencies from different evaluation models. With the calculated composite cross-efficiency scores provided in this study, DMUs are fully ranked accordingly. A measure for estimating the importance degree of cross-efficiency evaluation models is provided as well. The example of commercial banks in Taiwan is used to illustrate the approach proposed in this paper, and the results derived indicate that the proposed approach is able to provide composite cross-efficiency scores for ranking DMUs effectively.

## Figures and Tables

**Table 1 entropy-21-00467-t001:** Real data (in millions of Taiwan dollars) and CCR efficiency scores of 22 commercial banks in Taiwan.

Bank	Labor	Capital	Purchased Funds	Deposits	S-Term Loans	ML-Term Loans	CCR
1	9492	23,935	1,029,108	336,735	297,352	844,783	0.9327
2	848	2683	121,212	24,362	27,961	79,582	0.8496
3	2351	3416	323,449	106,247	104,348	259,497	1.0000
4	7306	14,299	815,246	279,769	339,261	617,217	1.0000
5	1388	2744	162,563	23,395	69,956	108,206	1.0000
6	1999	6195	125,917	15,016	30,227	69,487	0.6306
7	2838	7644	307,145	56,564	71,591	158,042	0.6205
8	3545	2814	325,073	48,824	48,539	247,323	0.9050
9	3585	3343	280,959	56,041	64,251	202,585	0.8058
10	1775	1128	204,472	21,517	36,705	150,177	0.9419
11	10,717	28,674	1,226,897	508,605	384,511	1,023,549	1.0000
12	9308	11,294	1,078,604	250,407	310,403	783,664	0.8654
13	9346	22,617	1,271,363	336,838	289,442	744,008	0.7767
14	6455	18,487	841,496	305,603	187,843	643,889	1.0000
15	3074	2150	395,750	66,537	92,533	307,930	1.0000
16	12,502	14,519	1,347,592	580,389	462,928	1,188,269	1.0000
17	9277	17,464	715,304	163,804	158,695	547,688	0.7958
18	3642	6915	505,286	105,395	103,643	341,020	0.8481
19	8049	11,002	616,242	232,732	218,083	594,174	1.0000
20	20,295	34,229	997,936	146,904	348,395	941,957	0.9831
21	11,405	27,730	1,243,848	476,748	404,671	1,028,704	0.9453
22	14,354	38,694	1,825,537	442,195	376,648	1,531,299	1.0000

**Table 2 entropy-21-00467-t002:** The cross-efficiency of 22 Taiwanese commercial banks via Doyle and Green [3].

Bank	1	2	3	4	5	6	7	8	9	10	11	12	13	14	15	16	17	18	19	20	21	22
1	0.9327	0.8076	0.9771	0.8680	0.7745	0.5105	0.5843	0.8173	0.7306	0.8502	0.9640	0.8427	0.7105	0.9183	0.9306	1.0000	0.7709	0.8238	0.9680	0.7764	0.9414	0.9997
2	0.8062	0.8496	1.0000	0.7653	0.7061	0.3149	0.5044	0.6320	0.5119	0.7662	0.8651	0.7627	0.7211	0.9036	0.9075	0.8610	0.5348	0.8481	0.6687	0.4204	0.8170	0.9663
3	0.9296	0.8248	1.0000	0.8850	0.8008	0.4866	0.5883	0.7890	0.7033	0.8428	0.9680	0.8489	0.7255	0.9249	0.9371	1.0000	0.7388	0.8352	0.9344	0.7172	0.9428	1.0000
4	0.8973	0.7476	0.9562	1.0000	0.9592	0.5835	0.6205	0.6781	0.7139	0.7215	0.9425	0.8392	0.6875	0.8077	0.8277	1.0000	0.7333	0.7282	1.0000	0.8732	0.9433	0.8381
5	0.8356	0.7796	1.0000	1.0000	1.0000	0.4385	0.5969	0.5470	0.5686	0.6585	0.9058	0.8164	0.7126	0.7929	0.8115	0.9414	0.5732	0.7368	0.8032	0.5735	0.8936	0.7977
6	0.8322	0.6649	0.8758	1.0000	0.9793	0.6306	0.6024	0.5872	0.6920	0.6218	0.8764	0.7863	0.6269	0.7041	0.7266	0.9454	0.7022	0.6338	1.0000	0.9831	0.8916	0.7122
7	0.8973	0.7476	0.9562	1.0000	0.9592	0.5835	0.6205	0.6781	0.7139	0.7215	0.9425	0.8392	0.6875	0.8077	0.8277	1.0000	0.7333	0.7282	1.0000	0.8732	0.9433	0.8381
8	0.7303	0.5957	0.9141	0.7474	0.6655	0.3391	0.4453	0.9050	0.7980	0.9419	0.7408	0.8295	0.5749	0.6963	1.0000	1.0000	0.6679	0.7193	0.9461	0.7053	0.7483	0.7738
9	0.7992	0.6475	0.9124	0.7882	0.6986	0.4152	0.4919	0.8876	0.8058	0.8950	0.8112	0.8273	0.6074	0.7560	0.9494	1.0000	0.7358	0.7371	1.0000	0.8168	0.8141	0.8362
10	0.7303	0.5957	0.9141	0.7473	0.6655	0.3391	0.4452	0.9050	0.7980	0.9419	0.7408	0.8295	0.5749	0.6962	1.0000	1.0000	0.6678	0.7193	0.9461	0.7053	0.7482	0.7738
11	0.8445	0.7384	1.0000	0.8484	0.5681	0.2850	0.5029	0.5211	0.4888	0.5500	1.0000	0.7123	0.7603	0.9871	0.7129	1.0000	0.5282	0.7460	0.7443	0.3570	0.9209	0.8462
12	0.5076	0.4353	1.0000	0.8981	0.9609	0.2191	0.3892	0.4681	0.5812	0.7379	0.5527	0.8654	0.5012	0.4191	0.9995	1.0000	0.3567	0.5417	0.7023	0.4043	0.5902	0.3966
13	0.7861	0.6859	1.0000	0.8379	0.4766	0.2056	0.4600	0.3754	0.3841	0.3837	1.0000	0.6378	0.7767	1.0000	0.5796	0.9832	0.4215	0.6850	0.6447	0.2257	0.8963	0.7391
14	0.8116	0.7115	1.0000	0.8342	0.5043	0.2351	0.4755	0.4418	0.4273	0.4605	1.0000	0.6687	0.7701	1.0000	0.6419	0.9891	0.4664	0.7164	0.6836	0.2757	0.9055	0.7934
15	0.8329	0.7377	1.0000	0.8168	0.7355	0.3773	0.5145	0.8210	0.7028	0.9020	0.8622	0.8506	0.6751	0.8366	1.0000	1.0000	0.6716	0.8166	0.8862	0.6256	0.8480	0.9163
16	0.9322	0.7890	0.9574	0.8654	0.7689	0.5381	0.5841	0.8295	0.7530	0.8453	0.9590	0.8374	0.6968	0.9036	0.9166	1.0000	0.7958	0.8066	1.0000	0.8408	0.9404	0.9857
17	0.9322	0.7890	0.9574	0.8654	0.7689	0.5381	0.5841	0.8295	0.7529	0.8454	0.9590	0.8374	0.6968	0.9036	0.9166	1.0000	0.7958	0.8066	1.0000	0.8407	0.9404	0.9857
18	0.8064	0.8496	1.0000	0.7654	0.7062	0.3151	0.5045	0.6322	0.5121	0.7663	0.8653	0.7628	0.7211	0.9036	0.9075	0.8611	0.5350	0.8481	0.6690	0.4207	0.8172	0.9664
19	0.9322	0.7890	0.9574	0.8654	0.7689	0.5381	0.5841	0.8295	0.7530	0.8453	0.9590	0.8374	0.6968	0.9036	0.9166	1.0000	0.7958	0.8066	1.0000	0.8408	0.9404	0.9857
20	0.8323	0.6650	0.8756	0.9988	0.9778	0.6303	0.6020	0.5883	0.6923	0.6226	0.8764	0.7862	0.6268	0.7046	0.7270	0.9452	0.7027	0.6342	1.0000	0.9831	0.8914	0.7131
21	0.9001	0.7889	1.0000	0.9999	0.9648	0.5175	0.6191	0.6638	0.6700	0.7369	0.9542	0.8508	0.7180	0.8416	0.8601	1.0000	0.6868	0.7667	0.9311	0.7310	0.9453	0.8720
22	0.9327	0.8080	0.9775	0.8680	0.7746	0.5100	0.5843	0.8170	0.7301	0.8503	0.9641	0.8428	0.7108	0.9186	0.9308	1.0000	0.7704	0.8241	0.9674	0.7752	0.9414	1.0000
Ave.	0.8382	0.7294	0.9651	0.8757	0.7811	0.4341	0.5411	0.6929	0.6583	0.7503	0.8959	0.8050	0.6809	0.8332	0.8649	0.9785	0.6538	0.7504	0.8861	0.6711	0.8755	0.8516

**Table 3 entropy-21-00467-t003:** The cross-efficiency of 22 Taiwanese commercial banks via Liang et al. [4].

Bank	1	2	3	4	5	6	7	8	9	10	11	12	13	14	15	16	17	18	19	20	21	22
1	0.9327	0.8076	0.9771	0.8680	0.7745	0.5105	0.5843	0.8173	0.7306	0.8502	0.9640	0.8427	0.7105	0.9183	0.9306	1.0000	0.7709	0.8238	0.9680	0.7764	0.9414	0.9997
2	0.8062	0.8496	1.0000	0.7653	0.7061	0.3149	0.5044	0.6320	0.5119	0.7662	0.8651	0.7627	0.7211	0.9036	0.9075	0.8610	0.5348	0.8481	0.6687	0.4204	0.8170	0.9663
3	0.9161	0.8083	1.0000	0.9375	0.8758	0.5005	0.6024	0.7321	0.6882	0.7944	0.9617	0.8497	0.7220	0.8867	0.9018	1.0000	0.7153	0.8037	0.9329	0.7234	0.9439	0.9413
4	0.8973	0.7476	0.9562	1.0000	0.9592	0.5835	0.6205	0.6781	0.7139	0.7215	0.9425	0.8392	0.6875	0.8077	0.8277	1.0000	0.7333	0.7282	1.0000	0.8732	0.9433	0.8381
5	0.8023	0.6800	0.8975	1.0000	1.0000	0.5265	0.5906	0.5205	0.6069	0.5924	0.8597	0.7763	0.6396	0.6978	0.7198	0.9167	0.6089	0.6379	0.8798	0.7391	0.8675	0.6937
6	0.8322	0.6649	0.8758	1.0000	0.9793	0.6306	0.6024	0.5872	0.6920	0.6218	0.8764	0.7863	0.6269	0.7041	0.7266	0.9454	0.7022	0.6338	1.0000	0.9831	0.8916	0.7122
7	0.8973	0.7476	0.9562	1.0000	0.9592	0.5835	0.6205	0.6781	0.7139	0.7215	0.9425	0.8392	0.6875	0.8077	0.8277	1.0000	0.7333	0.7282	1.0000	0.8732	0.9433	0.8381
8	0.7303	0.5957	0.9141	0.7474	0.6655	0.3391	0.4453	0.9050	0.7980	0.9419	0.7408	0.8295	0.5749	0.6963	1.0000	1.0000	0.6679	0.7193	0.9461	0.7053	0.7483	0.7738
9	0.7992	0.6475	0.9124	0.7882	0.6986	0.4152	0.4919	0.8876	0.8058	0.8950	0.8112	0.8273	0.6074	0.7560	0.9494	1.0000	0.7358	0.7371	1.0000	0.8168	0.8141	0.8362
10	0.7303	0.5957	0.9141	0.7473	0.6655	0.3391	0.4452	0.9050	0.7980	0.9419	0.7408	0.8295	0.5749	0.6962	1.0000	1.0000	0.6678	0.7193	0.9461	0.7053	0.7482	0.7738
11	0.8445	0.7384	1.0000	0.8484	0.5681	0.2850	0.5029	0.5211	0.4888	0.5500	1.0000	0.7123	0.7603	0.9871	0.7129	1.0000	0.5282	0.7460	0.7443	0.3570	0.9209	0.8462
12	0.5076	0.4353	1.0000	0.8981	0.9609	0.2191	0.3892	0.4681	0.5812	0.7379	0.5527	0.8654	0.5012	0.4191	0.9995	1.0000	0.3567	0.5417	0.7023	0.4043	0.5902	0.3966
13	0.7861	0.6859	1.0000	0.8379	0.4766	0.2056	0.4600	0.3754	0.3841	0.3837	1.0000	0.6378	0.7767	1.0000	0.5796	0.9832	0.4215	0.6850	0.6447	0.2257	0.8963	0.7391
14	0.8116	0.7115	1.0000	0.8342	0.5043	0.2351	0.4755	0.4418	0.4273	0.4605	1.0000	0.6687	0.7701	1.0000	0.6419	0.9891	0.4664	0.7164	0.6836	0.2757	0.9055	0.7934
15	0.8329	0.7377	1.0000	0.8168	0.7355	0.3773	0.5145	0.8210	0.7028	0.9020	0.8622	0.8506	0.6751	0.8366	1.0000	1.0000	0.6716	0.8166	0.8862	0.6256	0.8480	0.9163
16	0.9237	0.7787	0.9571	0.8983	0.8155	0.5489	0.5930	0.7928	0.7436	0.8150	0.9549	0.8378	0.6945	0.8800	0.8947	1.0000	0.7808	0.7873	1.0000	0.8483	0.9411	0.9494
17	0.9317	0.7882	0.9566	0.8648	0.7684	0.5383	0.5838	0.8293	0.7529	0.8448	0.9584	0.8368	0.6962	0.9029	0.9159	0.9995	0.7958	0.8059	1.0000	0.8414	0.9399	0.9850
18	0.8064	0.8496	1.0000	0.7654	0.7062	0.3151	0.5045	0.6322	0.5121	0.7663	0.8653	0.7628	0.7211	0.9036	0.9075	0.8611	0.5350	0.8481	0.6690	0.4207	0.8172	0.9664
19	0.9199	0.7732	0.9485	0.8870	0.8017	0.5490	0.5880	0.7969	0.7449	0.8150	0.9494	0.8321	0.6888	0.8769	0.8913	0.9944	0.7834	0.7835	1.0000	0.8544	0.9356	0.9483
20	0.8323	0.6650	0.8756	0.9988	0.9778	0.6303	0.6020	0.5883	0.6923	0.6226	0.8764	0.7862	0.6268	0.7046	0.7270	0.9452	0.7027	0.6342	1.0000	0.9831	0.8914	0.7131
21	0.9001	0.7889	1.0000	0.9999	0.9648	0.5175	0.6191	0.6638	0.6700	0.7369	0.9542	0.8508	0.7180	0.8416	0.8601	1.0000	0.6868	0.7667	0.9311	0.7310	0.9453	0.8720
22	0.9327	0.8080	0.9775	0.8680	0.7746	0.5100	0.5843	0.8170	0.7301	0.8503	0.9641	0.8428	0.7108	0.9186	0.9308	1.0000	0.7704	0.8241	0.9674	0.7752	0.9414	1.0000
Ave.	0.8352	0.7230	0.9599	0.8805	0.7881	0.4398	0.5420	0.6859	0.6586	0.7424	0.8928	0.8030	0.6769	0.8248	0.8569	0.9771	0.6531	0.7425	0.8896	0.6799	0.8742	0.8409

**Table 4 entropy-21-00467-t004:** The cross-efficiency of 22 Taiwanese commercial banks via Liang et al. [5].

Bank	1	2	3	4	5	6	7	8	9	10	11	12	13	14	15	16	17	18	19	20	21	22
1	0.9327	0.8080	0.9776	0.8681	0.7747	0.5105	0.5843	0.8173	0.7306	0.8504	0.9641	0.8428	0.7109	0.9186	0.9309	1.0000	0.7709	0.8242	0.9680	0.7764	0.9414	1.0000
2	0.8062	0.8496	1.0000	0.7653	0.7061	0.3149	0.5044	0.6320	0.5119	0.7662	0.8651	0.7627	0.7211	0.9036	0.9075	0.8610	0.5348	0.8481	0.6687	0.4204	0.8170	0.9663
3	0.9296	0.8496	1.0000	1.0000	1.0000	0.5175	0.6192	0.8210	0.7066	0.9020	1.0000	0.8654	0.7767	1.0000	1.0000	1.0000	0.7388	0.8481	0.9344	0.7311	0.9453	1.0000
4	0.9000	0.7889	1.0000	1.0000	1.0000	0.6306	0.6205	0.6781	0.7262	0.7368	0.9839	0.8541	0.7520	0.8731	0.8617	1.0000	0.7333	0.7666	1.0000	0.9831	0.9453	0.8719
5	0.8356	0.7796	1.0000	1.0000	1.0000	0.6103	0.5969	0.5992	0.7076	0.7108	0.9058	0.8637	0.7132	0.7929	0.9405	1.0000	0.6384	0.7368	0.9617	0.9287	0.8936	0.7977
6	0.8322	0.6649	0.8758	1.0000	0.9793	0.6306	0.6024	0.5872	0.6920	0.6218	0.8764	0.7863	0.6269	0.7041	0.7266	0.9454	0.7022	0.6338	1.0000	0.9831	0.8916	0.7122
7	0.8973	0.7477	0.9563	1.0000	0.9592	0.5835	0.6205	0.6781	0.7139	0.7215	0.9425	0.8392	0.6875	0.8077	0.8277	1.0000	0.7333	0.7283	1.0000	0.8732	0.9433	0.8382
8	0.7303	0.5957	0.9141	0.7474	0.6655	0.3391	0.4453	0.9050	0.7980	0.9419	0.7408	0.8295	0.5749	0.6963	1.0000	1.0000	0.6679	0.7193	0.9461	0.7053	0.7483	0.7738
9	0.7992	0.6475	0.9124	0.7882	0.6986	0.4152	0.4919	0.8877	0.8058	0.8951	0.8112	0.8273	0.6074	0.7560	0.9494	1.0000	0.7358	0.7371	1.0000	0.8168	0.8141	0.8362
10	0.7303	0.5957	0.9141	0.7473	0.6655	0.3391	0.4452	0.9050	0.7980	0.9419	0.7408	0.8295	0.5749	0.6962	1.0000	1.0000	0.6678	0.7193	0.9461	0.7053	0.7482	0.7738
11	0.8445	0.7384	1.0000	0.9372	0.6270	0.2850	0.5029	0.5211	0.4888	0.5500	1.0000	0.7123	0.7767	1.0000	0.7129	1.0000	0.5282	0.7460	0.7443	0.3570	0.9209	0.8462
12	0.5076	0.4353	1.0000	0.8981	0.9609	0.2191	0.3892	0.4685	0.5813	0.7384	0.5527	0.8654	0.5012	0.4191	1.0000	1.0000	0.3567	0.5418	0.7023	0.4043	0.5902	0.3967
13	0.7861	0.6859	1.0000	0.8883	0.5305	0.2056	0.4673	0.3754	0.3841	0.3837	1.0000	0.6378	0.7767	1.0000	0.5796	0.9873	0.4215	0.6850	0.6447	0.2257	0.8989	0.7391
14	0.8116	0.7115	1.0000	0.8379	0.5043	0.2351	0.4755	0.4418	0.4273	0.4605	1.0000	0.6687	0.7767	1.0000	0.6419	1.0000	0.4664	0.7164	0.6836	0.2757	0.9055	0.7934
15	0.8329	0.7377	1.0000	0.8978	0.9606	0.3773	0.5145	0.9050	0.7980	0.9419	0.8622	0.8654	0.6751	0.8366	1.0000	1.0000	0.6716	0.8166	0.9461	0.7053	0.8480	0.9163
16	0.9327	0.8248	1.0000	1.0000	1.0000	0.5835	0.6205	0.9050	0.8058	0.9419	1.0000	0.8654	0.7743	1.0000	1.0000	1.0000	0.7958	0.8397	1.0000	0.8732	0.9453	1.0000
17	0.9322	0.7890	0.9574	0.8654	0.7689	0.5383	0.5841	0.8295	0.7530	0.8454	0.9590	0.8374	0.6968	0.9036	0.9166	1.0000	0.7958	0.8066	1.0000	0.8414	0.9404	0.9857
18	0.8064	0.8496	1.0000	0.7654	0.7062	0.3151	0.5045	0.6322	0.5121	0.7663	0.8653	0.7628	0.7211	0.9036	0.9075	0.8611	0.5350	0.8481	0.6690	0.4207	0.8172	0.9664
19	0.9322	0.7890	0.9574	1.0000	0.9835	0.6306	0.6205	0.8876	0.8058	0.8950	0.9590	0.8392	0.6968	0.9036	0.9494	1.0000	0.7958	0.8066	1.0000	0.9831	0.9433	0.9857
20	0.8323	0.6650	0.8758	1.0000	0.9793	0.6306	0.6024	0.5883	0.6923	0.6226	0.8764	0.7863	0.6269	0.7046	0.7270	0.9454	0.7027	0.6342	1.0000	0.9831	0.8916	0.7131
21	0.9001	0.7889	1.0000	1.0000	0.9649	0.5176	0.6192	0.6638	0.6700	0.7369	0.9542	0.8508	0.7180	0.8416	0.8601	1.0000	0.6868	0.7667	0.9311	0.7312	0.9453	0.8720
22	0.9327	0.8354	1.0000	0.8850	0.8008	0.5100	0.5883	0.8170	0.7301	0.8622	0.9726	0.8489	0.7295	0.9399	0.9539	1.0000	0.7704	0.8450	0.9674	0.7752	0.9428	1.0000
Ave.	0.8384	0.7354	0.9700	0.9041	0.8289	0.4518	0.5463	0.7066	0.6745	0.7651	0.9014	0.8110	0.6916	0.8455	0.8815	0.9818	0.6568	0.7552	0.8961	0.7045	0.8763	0.8538

**Table 5 entropy-21-00467-t005:** Three different cross-efficiency scores for 22 commercial banks in Taiwan.

Bank	Model (5)	Model (8)	Model (9)
1	0.8382 (9)	0.8352 (9)	0.8384 (10)
2	0.7294 (15)	0.7230 (15)	0.7354 (15)
3	0.9651 (2)	0.9599 (2)	0.9700 (2)
4	0.8757 (5)	0.8805 (5)	0.9041 (3)
5	0.7811 (12)	0.7881 (12)	0.8289 (11)
6	0.4341 (22)	0.4398 (22)	0.4518 (22)
7	0.5411 (21)	0.5420 (21)	0.5463 (21)
8	0.6929 (16)	0.6859 (16)	0.7066 (16)
9	0.6583 (19)	0.6586 (19)	0.6745 (19)
10	0.7503 (14)	0.7424 (14)	0.7651 (13)
11	0.8959 (3)	0.8928 (3)	0.9014 (4)
12	0.8050 (11)	0.8030 (11)	0.8110 (12)
13	0.6809 (17)	0.6769 (18)	0.6916 (18)
14	0.8332 (10)	0.8248 (10)	0.8455 (9)
15	0.8649 (7)	0.8569 (7)	0.8815 (6)
16	0.9785 (1)	0.9771 (1)	0.9818 (1)
17	0.6538 (20)	0.6531 (20)	0.6568 (20)
18	0.7504 (13)	0.7425 (13)	0.7552 (14)
19	0.8861 (4)	0.8896 (4)	0.8961 (5)
20	0.6711 (18)	0.6799 (18)	0.7045 (17)
21	0.8755 (6)	0.8742 (6)	0.8763 (7)
22	0.8516 (8)	0.8409 (8)	0.8538 (8)

**Table 6 entropy-21-00467-t006:** Normalized and composite cross-efficiency scores for 22 commercial banks in Taiwan.

Bank	Model (5)	Model (8)	Model (9)	$E_{j}^{C*}$	Rank
1	0.0493	0.0492	0.0485	0.8373	9
2	0.0429	0.0426	0.0426	0.7292	15
3	0.0567	0.0566	0.0561	0.9650	2
4	0.0515	0.0519	0.0523	0.8865	5
5	0.0459	0.0464	0.0480	0.7989	12
6	0.0255	0.0259	0.0261	0.4417	22
7	0.0318	0.0319	0.0316	0.5431	21
8	0.0407	0.0404	0.0409	0.6950	16
9	0.0387	0.0388	0.0390	0.6636	19
10	0.0441	0.0438	0.0443	0.7525	13
11	0.0527	0.0526	0.0522	0.8966	3
12	0.0473	0.0473	0.0469	0.8063	11
13	0.0400	0.0399	0.0400	0.6830	18
14	0.0490	0.0486	0.0489	0.8344	10
15	0.0508	0.0505	0.0510	0.8676	7
16	0.0575	0.0576	0.0568	0.9791	1
17	0.0384	0.0385	0.0380	0.6546	20
18	0.0441	0.0438	0.0437	0.7493	14
19	0.0521	0.0524	0.0519	0.8905	4
20	0.0394	0.0401	0.0408	0.6849	17
21	0.0515	0.0515	0.0507	0.8753	6
22	0.0501	0.0496	0.0494	0.8487	8

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
