# Peer review of "Ranking DMUs by Combining Cross-Efficiency Scores Based on Shannon’s Entropy"

_entropy, 2019, doi:10.3390/e21050467_

Round 1

Reviewer 1 Report

The topic is very actual and interesting. Paper is focused on cross-efficiency. However, I have several suggestions. The aim of the paper is not obvious from the Introduction. I recommend to extend the literature review and discuss the findings with the previous studies. The methodology is well described and presented. I suggest to better describe the example. The data set and selection of inputs and outpus are not described enough. The results are very similar and I recommend to extend and better present the three cross-efficiency scores. Where there are differences in results of different model? How can be these findings used in practice? The policy implication is missing. I recommend to better present the findings and results and discuss the results.

Author Response

The general comment of this reviewer is:

The topic is very actual and interesting. Paper is focused on cross-efficiency. However, I have several suggestions.

We thank for this general comment, and address the issues raised by the reviewer in the followings.

1. The aim of the paper is not obvious from the Introduction. I recommend to extend the literature review and discuss the findings with the previous studies.

In response to this comment, we rewrite the Introduction for describing the purpose of this paper more clearly, and extend the literature review. However, since this paper focuses on the application of Shannon’s entropy to rank decision making units, we do not intend to discuss the findings with previous studies.

2. The methodology is well described and presented. I suggest to better describe the example. The data set and selection of inputs and outputs are not described enough.

We borrow the data set from Liu [10] to explain the idea proposed in this paper, and the data set and selection of input and output items are described clearly in Liu [10].

3. The results are very similar and I recommend to extend and better present the three cross-efficiency scores. Where there are differences in results of different model? How can be these findings used in practice?

Different ways for determining the weights produce different cross efficiencies, and lead to different ranking results of DMUs. Nevertheless, these models may provide some precious information that we cannot ignore. For this reason, we calculate Shannon’s entropy of the obtained cross-efficiencies from several cross-efficiency models for ranking DMUs. In practice, we need several cross-efficiency evaluation models to simultaneously measuring the cross-efficiency scores of DMUs and combine their derived results for making decision.

4. The policy implication is missing. I recommend to better present the findings and results and discuss the results.

Of the 22 commercial banks, based on their calculated composite cross-efficiencies, bank no. 16 is in first place, followed by banks no. 3, 11, 19, and 4 subsequently. This shows that Combining the efficiency results of different cross-efficiency models give more practical ranking results for decision makers compared with using each of cross-efficiency models respectively. For example, if a DMU has a strict maximum output value for any output item, then it may be measured as an efficient unit by some cross-efficiency evaluation models. Under this situation the model might decide the efficiency based upon one factor, and it seems to be not reasonable. This can be avoided by using the procedure for the combination of different cross-efficiencies proposed in this study. This is elaborated on page 11 in the revision.

Reviewer 2 Report

It is a very good paper, clearly written and of interest form the DEA community.

My suggestion for improving the paper is to provide information regarding the importance of the degree of each cros -efficiency model.

Indeed, it can show which model fits well with your data set. It can also provide insight on the weight of each model in the final DEA score; thus supporting a better decision-making as well as a classification of the most commonly used cross-efficiency models.

Author Response

The general comment of this reviewer is:

It is a very good paper, clearly written and of interest form the DEA community.  

We thank for the encouragement of the reviewer. This reviewer has only one comment, and the issue given by the reviewer is responded in the followings.

1.      My suggestion for improving the paper is to provide information regarding the importance of the degree of each cross -efficiency model. Indeed, it can show which model fits well with your data set. It can also provide insight on the weight of each model in the final DEA score; thus supporting a better decision-making as well as a classification of the most commonly used cross-efficiency models.

To responded to this comment, the degree of importance for each model is calculated as w1=0.3419, w2=0.3348, and w3=0.3233, respectively. This is stressed at the first paragraph on page of the revision.